# A Comparative Assessment of Approvals and Discontinuations of Systemic Antibiotics and Other Therapeutic Areas

**DOI:** 10.3390/healthcare11121759

**Published:** 2023-06-15

**Authors:** Rosa Rodriguez-Monguio, Enrique Seoane-Vazquez, John H. Powers

**Affiliations:** 1Department of Clinical Pharmacy, School of Pharmacy, University of California San Francisco, San Francisco, CA 93143-0622, USA; 2Medication Outcomes Center, University of California San Francisco, San Francisco, CA 93143-0622, USA; 3Philip R. Lee Institute for Health Policy Studies, University of California San Francisco, San Francisco, CA 93143-0622, USA; 4Department of Biomedical and Pharmaceutical Sciences, Chapman University School of Pharmacy, Irvine, CA 92618, USA; seoanevazquez@chapman.edu; 5Economics Science Institute, Argyros School of Business and Economics, Chapman University, Orange, CA 92866, USA; 6Department of Medicine, George Washington University School of Medicine, Washington, DC 20052, USA; jpowers3@aol.com

**Keywords:** infectious diseases, anti-infectives, antibiotics, drug-resistant pathogens, patient outcomes

## Abstract

Since 1980, the US Congress has passed legislation providing several incentives to encourage the development and regulatory approval of new drugs, particularly antibiotics. We assessed long-term trends and characteristics of approvals and discontinuations of all new molecular entities, new therapeutic biologics, and gene and cell therapies approved by the US Food and Drug Administration (FDA), as well as reasons for discontinuations by therapeutic class, in the context of laws and regulations implemented over the past four decades. In the period 1980–2021, the FDA approved 1310 new drugs, of which 210 (16.0%) had been discontinued as of 31 December 2021, including 38 (2.9%) withdrawn for safety reasons. The FDA approved 77 (5.9%) new systemic antibiotics, of which 32 (41.6%) had been discontinued at the end of the observation period, including 6 (7.8%) safety withdrawals. Since the enactment of the FDA Safety and Innovation Act in 2012, which created the Qualified Infectious Disease Product designation for antiinfectives to treat serious or life-threatening diseases due to resistant or potentially resistant bacteria, the FDA has approved 15 new systemic antibiotics, all using non-inferiority trials, for 22 indications and five different infections. Only one of the infections had labeled indications for patients with drug-resistant pathogens.

## 1. Introduction

Over the past 40 years, the US Congress has enacted laws aimed at accelerating the development and speeding the regulatory review and approval of pharmaceuticals in general and new systemic antibiotics in particular. In 1992, the Prescription Drug User Fee Act (PDUFA) authorized the Food and Drug Administration (FDA) to collect fees from pharmaceutical companies to shorten the FDA’s regulatory review time for new drugs, including systemic antibiotics. In 1997, the FDA Modernization Act (FDAMA) extended drug market exclusivities and patent extensions to antibiotics to encourage the development of new systemic antibiotics. In 2012, the Generating Antibiotic Incentives Now (GAIN) Act was passed as part of the Food and Drug Administration Safety and Innovation Act (FDASIA). FDASIA (2012) amended the accelerated approval pathway and the fast track and breakthrough therapy designations for eligible new antibacterial and antifungal drugs intended to treat serious or life-threatening infections. FDASIA also created the qualifying infectious disease product (QIDP) designation for anti-infectives, speeding review times for drugs intended to treat serious or life-threatening infections caused by antibacterial or antifungal resistant pathogens. This law directed the FDA to create a list of qualifying pathogens that have the potential to pose a serious threat to public health and added five years of market exclusivity, in addition to any other exclusivity periods applicable under the Hatch–Waxman Act, for qualified infectious disease products without requiring evidence of improved patient outcomes [1]. More recently, the 21st Century Cures Act (2016) established the limited population antimicrobial drug regulatory pathway for treatment of a serious or life-threatening infection in a limited population of patients with unmet needs and required the FDA to create a system to expedite the recognition of antimicrobial susceptibility test interpretive criteria to determine antibiotic in vitro susceptibility [2].

Drug sponsor companies may discontinue approved drugs from the market at any time without warning. The only exception to drug market discontinuations is in cases of a potential disruption in the US market supply, in which case companies must notify the FDA of drug discontinuations at least six months in advance. Likewise, the FDA may withdraw approval of a drug application or an approved drug at any given time due to safety or efficacy-related reasons [3]. According to the FDA, a drug market discontinuation is a situation in which a drug is no longer distributed commercially by an FDA-regulated manufacturer, irrespective of whether the manufacturer makes a request for the formal withdrawal of the drug approval application or an abbreviated drug application to the FDA [4].

Prior studies evaluating the approval and discontinuation of antibiotic drugs were constrained by limited timeframes and did not assess the impact of major regulatory actions passed in the United States [5,6,7]. Moreover, prior studies did not compare antibiotic approvals with those of other therapeutic classes. Hence, this study assessed long-term trends and characteristics of FDA approved new systemic antibiotics and discontinuations, evaluated reasons associated with those market discontinuations, and compared systemic antibiotic approvals and discontinuations with all other therapeutic classes in the context of laws enacted and regulations implemented in the US over the last four decades. Study findings may inform legal and regulatory systems as well as public health programs and advance understanding of the effectiveness of enacted laws and implemented policies bringing more and novel systemic antibiotics to the US market compared to available therapeutic alternatives.

## 2. Materials and Methods

This observational study included all new molecular entities (NME—a new drug containing an active ingredient that has never been approved before for marketing in the US), new therapeutic biologics regulated by the FDA Centers for Drug Evaluation and Research (CDER), and gene and cell therapies regulated by the Centers for Biologic Evaluation and Research (CBER), approved by the FDA in the period 1980–2021.

We extracted data for each FDA-approved NME, biologic license application, and gene and cell therapy, including the new drug application number, filling and approval dates, product number and type of application, active ingredient, generic and brand names, dosage form and administration route, National Drug Code, and market status. We also collected information on the FDA approval pathway (priority review, standard review) and orphan drug designation. NMEs and therapeutic biologics data were derived from the Orange Book, Drugs@FDA, and Purple Book, respectively. Gene and cell therapy data were derived from the list of approved cellular and gene therapy products approved in the US available at the CBER [8,9,10,11]. We also compiled information about notices of FDA regulatory actions from the Federal Register [12]. We extracted the therapeutic classification from the World Health Organization Collaborating Center for Drug Statistics Methodology Anatomical Therapeutic Chemical (ATC) Classification System [13]. We excluded vaccines or other biological products authorized by the CBER because of missing historical information. We extracted all data through 31 December 2021.

We collected study data according to a predetermined protocol using a standardized data extraction form developed and piloted by the authors and used in prior studies [14]. To ensure the reliability of the data collection, one study investigator (ES) was responsible for primary data extraction and placement into evidence tables. A second investigator (RRM) verified the data extraction and entry processes. We discussed discrepancies with a third investigator (JP) and solved them by consensus.

A drug was defined as discontinued from the US market as of 31 December 2021, if the FDA had released a notice of discontinuation of approval in the Federal Register, the drug was listed as discontinued in the market status section of the Drug@FDA database, the drug was no longer included in the Orange Book (OB) listing of approved applications, or it was listed in the OB discontinuation section. Drug withdrawal data for safety reasons were derived from the Federal Register, the FDA Safety Information and Adverse Event Reporting Program (MedWatch), and other safety communications available at the FDA webpage [12,15,16].

The unit of analysis was the first new drug application or biologic license application for each NME, new therapeutic biologic, and gene and cell therapy approved by the FDA during the study period. We assessed trends over time in approvals and discontinuations of systemic antibiotics and all other new drugs approved by the FDA during the period by four distinct regulatory periods (i.e., PDUFA, FDAMA, FDASIA, and the 21st Century Cures Act) demarcated by the enactment of major regulatory actions.

We conducted descriptive analyses for study variables including approvals, discontinuations, and safety withdrawals. We used the non-parametric Pearson’s chi-squared test of independence to assess differences in market discontinuations and safety withdrawals among therapeutic categories. The significance level was set at 0.05. All analyses were performed using R.

## 3. Results

In the period 1980–2021, the FDA approved 1310 new drugs. The therapeutic class with the greatest number of approvals was antineoplastic drugs (285; 21.8% of all approvals), followed by the central nervous system (154; 11.8%), and the alimentary tract and metabolism (129; 9.8%). Approvals of antineoplastic and immunomodulating agents increased steadily over time from an annual average of 2.0 drugs (26; 8.8%) prior to PDUFA to 19.5 (98; 36.7%) after the enactment of the 21st Century Cures Act (Table 1). Conversely, approvals of cardiovascular drugs decreased from an annual average of 4.1 drugs (54; 18.2%) prior to PDUFA to 1.6 (8; 3.0%) in the period 2016–2021 (Figure 1 and Figure 2).

Overall, 210 drugs, representing 16.0% of all FDA-approved drugs during the study period, had been discontinued from the market as of 31 December 2021 (Table 2). Market discontinuations were significantly greater for antibiotics for systemic use (32; 41.6% of all approved systemic antibiotics) and diagnostic drugs (32; 38.1%) than any other therapeutic class (*p* < 0.05) except for diagnostic drugs (*p* = 0.690) and antiparasitic products, insecticides, and repellents (*p* = 0.273) (Figure 3). The FDA withdrew 38 drugs (18.3% of all discontinued drugs) for safety reasons, including six (15.8% of safety discontinuations) systemic antibiotics (Table 2). Discontinuation for safety reasons was significantly greater for antibacterials for systemic use than for antineoplastic and immunomodulating agents (*p* = 0.004), the nervous system (*p* = 0.025), diagnostic drugs (*p* = 0.037), and antivirals for systemic use (*p* = 0.021). All discontinued systemic antibiotics were approved prior to the FDASIA in 2012. Systemic antibiotic discontinuations occurred in pharmacology classes with a large number of therapeutic alternatives, including quinolone antibacterials (11 discontinuations, 73.3% of the drugs in this class) and other beta-lactam antibacterials (13; 39.4%) (Table 3).

In the study period, the FDA approved 77 antibiotics for systemic use (Appendix A). FDA Approved New Systemic Antibiotics by Pharmacology Class, 1980–2021). The FDA approved on average 3.0 antibiotics yearly (12.8% of all new drug approvals) in the period 1980-PDUFA, 1.6 (4.5%) in PDUFA-FDAMA, 1.1 (3.9%) in FDAMA- FDASIA, 1.1 (3.1%) in FDASIA-21st Century Cures Act, and 2.0 (3.7%) after the enactment of the 21st Century Cures Act (Table 1). Since the enactment of FDASIA (2012), the FDA has approved 15 new antibiotics for systemic use and granted the QIDP designation to 14 of them. Those 14 QIDP antibiotics, except cefiderocol and meropenem/vaborbactam, were already under clinical development prior to the enactment of FDASIA.

Out of 77 new antibiotics for systemic use approved by the FDA in the study period, 50 new antibiotics were indicated for skin and skin structure infections, 48 for urinary tract infections, and 34 were for pneumonia (Appendix A. Indications of FDA Approved New Antibiotics, 1980–2021). These indications were tested and approved based on studies in patients infected with bacteria susceptible to the antibiotic used as a comparator in the clinical trial.

After the enactment of FDASIA in 2012, the FDA approved 15 new antibiotics for systemic use and 22 indications targeting five different infections (skin and skin structure infections, urinary tract infections, intra-abdominal infections, and community- and hospital-acquired pneumonia). Five systemic antibiotics were indicated for bacterial skin and skin structure infections, all of them due to methicillin-resistant Staphylococcus aureus (MRSA).

## 4. Discussion

In this study assessing FDA approvals and discontinuations of pharmaceutical products since 1980, we found an overall upward trend in the number of approvals. The therapeutic classes with the greatest number of approvals were antineoplastic, which in most recent years increased to one in three approved drugs, followed by central nervous system, alimentary tract, and metabolism drugs. These findings are consistent with the trend identified in previous studies conducted during the 2010s and over shorter timeframes [5,6]. We also found that the average number of systemic antibiotics approved per year was higher in the 1980s and declined after the enactment of PDUFA. However, prior to PDUFA, many systemic antibiotics were approved for non-serious, self-resolving diseases such as acute otitis media, sinusitis, and bronchitis. FDAMA’s expansion of market exclusivities and patent extensions for antibiotics did not reverse the overall downward trend in the approvals of new systemic antibiotics. Approvals of new systemic antibiotics, as a percentage of total approvals, increased after the enactment of FDASIA. Yet, except for cefiderocol and meropenem/vaborbactam, new systemic antibiotics were already under development prior to the enactment of FDASIA, and none of them have demonstrated improved patient outcomes when compared to older, less expensive alternatives [6,17]. Approvals of new systemic antibiotics concentrated on pharmacology classes with large numbers of therapeutic alternatives and relatively few infections, namely skin and skin structure infections, urinary tract infections, intra-abdominal infections, pneumonia, and infections due to bacteria susceptible to studied drugs.

In the study period, antibiotics for systemic use represented the sixth therapeutic class in number of approvals but the first class in market discontinuations. The pharmacology classes with the greatest number of discontinuations were antibacterials for systemic use, diagnostic drugs, and antivirals. Almost one in five discontinuations were due to safety concerns.

Congressional legislation enacted over the past 40 years to encourage the development and regulatory approval of new drugs was intended to address the need for greater effectiveness of new interventions in patients who lack effective options. The purpose of the regulations implemented to encourage the development of those new drugs was to offer improved outcomes for patients with unmet medical needs. Yet, newly developed systemic antibiotics did not provide evidence of greater efficacy relative to the drugs to which they were compared, or improved patient outcomes compared to older, less expensive generic alternatives. This is particularly the case for the QIDP designation for anti-infectives to treat serious or life-threatening diseases since the law and regulation do not require evidence of improved patient outcomes, a necessary requirement to address unmet needs for greater effectiveness. Furthermore, new antibiotics for systemic use granted the QIDP designation were evaluated for regulatory approval using non-inferiority clinical trials, and the patients enrolled in those trials had bacterial infections susceptible to already available therapeutic alternatives, thus potentially allowing for worse effectiveness compared to available therapies. This is evident in the case of antibiotics for complicated urinary tract infections (cUTI). While there were three dozen systemic antibiotics for cUTI marketed in the US, none of the six QIDP-designated antibiotics approved for cUTI demonstrated evidence of improved patient outcomes (clinical response at the test of cure) compared to existing therapeutic options. Moreover, data derived from UTI studies is complex to interpret due to the utilization of urine cultures as a surrogate outcome whose relationship to direct patient outcomes of survival, symptoms, and patient function remains unclear. Lastly, FDASIA’s intended goal was to encourage the development of anti-infectives to treat serious or life-threatening diseases caused by resistant or potentially resistant bacteria, meaning patients who lack effective options. MRSA was the only resistant bacteria targeted by the antibiotics approved with the QIDP designation.

During the past four decades, a large proportion of antibiotics have been discontinued from the US market, particularly when compared to other therapeutic areas. More specifically, market discontinuations of systemic antibiotics occurred at three times the rate of discontinuation in other drug classes. Furthermore, two in five newly approved systemic antibiotics were subsequently withdrawn from the market, and those discontinuations concentrated in pharmacological classes with a large number of therapeutic alternatives. While drug market discontinuations are correlated with the number of years after market entry, overall, drug discontinuations occurred after the market entry of therapeutic alternatives with more favorable risk–benefit trade-offs [14]. This was not the case for systemic antibiotics—one in five of the systemic antibiotics withdrawn from the market were due to safety related concerns, and none of them were attributed to the emergence of drug resistant bacteria. Discontinuations due to potential harms in drugs studied in non-inferiority trials should be uncommon enough to warrant the potential loss of efficacy.

Pharmaceutical companies may also discontinue products with low utilization due to financial considerations and market-related strategies. Prior studies found that the usage of new antibiotics was lower than anticipated due to high prices and a lack of substantial evidence from clinical trials demonstrating superior efficacy or safety [18,19,20,21]. One study found that the prices of bedaquiline (2012), dalbavancin (2014), tedizolid (2014), oritavancin (2014), ceftolozane–tazobactam (2014), and ceftazidime–avibactam (2014) were significantly higher than the comparator antibiotic used in clinical trials despite lacking evidence of improving patient outcomes [6]. Another study estimated that the cost of a 14-day treatment course of ceftaroline for MRSA would be six times higher than the same course of treatment with daptomycin and sixty times higher than the cost of a 14-day treatment course of vancomycin, the current first-line treatment for these infections [20]. Similarly, the cost of 14-day treatment with cefiderocol for Gram-negative infections (carbapenem-resistant *Acinetobacter baumannii*, CRAB) was estimated to be 20 times higher than that of colistimethate. The randomized clinical trial comparing both antibiotics showed that cefiderocol was associated with increased all-cause mortality, particularly in CRAB infections [22,23]. In the absence of evidence of clinically meaningful added benefits, low uptake of new antibiotics in clinical practice may serve as a gauge of specific antibiotics’ perceived therapeutic value in addressing unmet medical needs. In this regard, the low utilization of pricey antibiotics that lack evidence of clinical superiority compared to available therapeutic options as well as the subsequent discontinuation from their market are indicative of typical market behavior.

Lastly, short-term treatment courses, compared to drugs indicated for chronic conditions, have been blamed for the marginal development of new antibiotics and the high market discontinuation rate [24]. Nevertheless, in this study, and others before [5], we found an overall downward trend in the approval of systemic antibiotics comparable to the trend observed for new drugs for chronic, prevalent conditions in the US, including cardiovascular and musculoskeletal system drugs.

This study has some limitations. The data included the first NDA for an NME. An NME or new therapeutic biologic does not necessarily add patient benefits compared to available therapies. Conversely, a new approval of a variation of an antibiotic already marketed may represent an improvement over the available alternatives if it is shown to improve patient outcomes. This study included systemic antibiotics and excluded topical antibiotics. Drug sponsor companies may not provide reasoning for market discontinuations; hence, discontinuations due to safety concerns may be underestimated. Study findings need to be interpreted in the context of regulatory changes implemented during the study period and the number of years an antibiotic has been on the market. Future research could compare the regulatory approval pathways, designations, and indications of systemic antibiotics approved by the FDA and the European Medicines Agency.

## 5. Conclusions

Regulatory and financial incentives to accelerate the development and regulatory review of new drugs resulted in increased approvals of systemic antibiotics. The discontinuation rate of systemic antibiotics was greater than any other therapeutic class, more prevalent in pharmacological classes with a greater number of therapeutic alternatives, and not related to safety or antibacterial resistance concerns. QIDP-designated antibiotics were approved based on non-inferiority trials for indications with multiple marketed antibiotics and infections due to pathogens susceptible to currently available drugs. Over the past four decades, legislation and subsequent regulations have led to an increased availability of systemic antibiotics. However, there remains a critical need for the development of innovative therapies for patients with susceptible and resistant pathogens who lack effective options to enhance their survival, improve functioning, and reduce their burden of symptoms.

## Figures and Tables

**Figure 1 healthcare-11-01759-f001:**
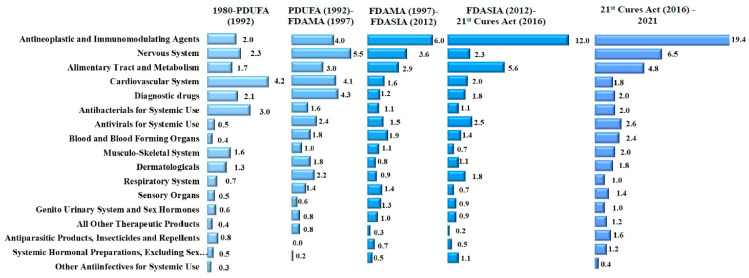
Average annual new drug approvals by therapeutic class and regulatory period, 1980–2021.

**Figure 2 healthcare-11-01759-f002:**
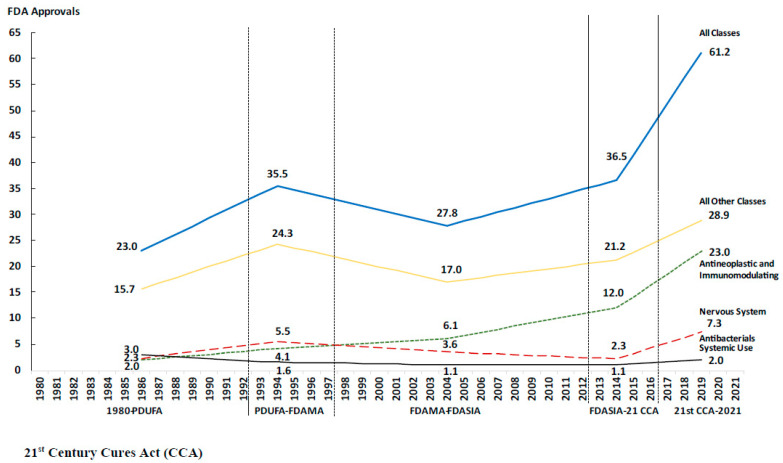
Trends in average annual number of approvals by therapeutic class and regulatory period.

**Figure 3 healthcare-11-01759-f003:**
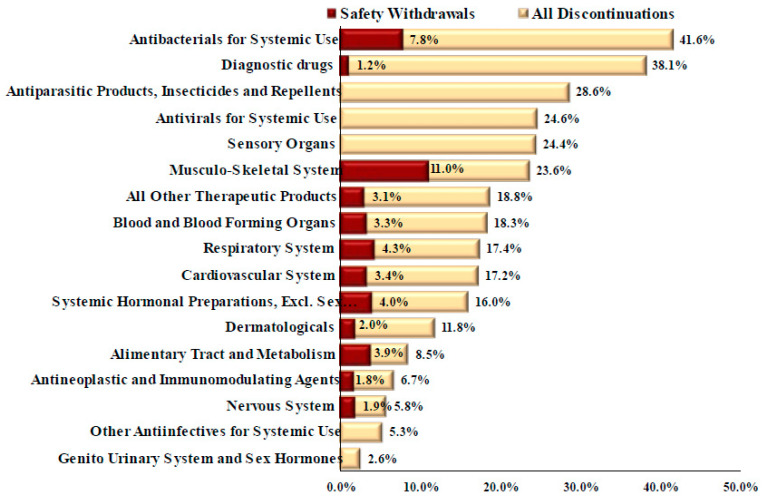
Market discontinuations and safety withdrawals of FDA new drug approvals by therapeutic class, 1980–2021.

**Table 1 healthcare-11-01759-t001:** FDA-approved new drugs by therapeutic class, 1980–2021.

Therapeutic Class	1980–PDUFA (1992)	PDUFA (1992)–FDAMA (1997)	FDAMA (1997)–FDASIA (2012)	FDASIA (2012)–21st Century Cures Act (2016)	21st Century Cures Act (2016)–2021	Total
**Antineoplastic and Immunomodulating Agents**	26 (2.0, 8.8%)	20 (4.0, 11.2%)	88 (6.0, 21.7%)	53 (12.0, 32.7%)	98 (19.5, 36.7%)	285 (6.8, 21.8%)
**Nervous System**	30 (2.3, 10.1%)	28 (5.5, 15.6%)	52 (3.6, 12.8%)	10 (2.3, 6.2%)	34 (6.8, 12.7%)	154 (3.7, 11.8%)
**Alimentary Tract and Metabolism**	22 (1.7, 7.4%)	15 (3.0, 8.4%)	43 (2.9, 10.6%)	25 (5.6, 15.4%)	24 (4.8, 9.0%)	129 (3.1, 9.8%)
**Cardiovascular System**	54 (4.2, 18.2%)	21 (4.1, 11.7%)	24 (1.6, 5.9%)	9 (2.0, 5.6%)	8 (1.6, 3.0%)	116 (2.8, 8.9%)
**Diagnostic drugs**	27 (2.1, 9.1%)	22 (4.3, 12.3%)	17 (1.2, 4.2%)	8 (1.8, 4.9%)	10 (2.0, 3.7%)	84 (2.0, 6.4%)
**Antibacterials for Systemic Use**	38 (3.0, 12.8%)	8 (1.6, 4.5%)	16 (1.1, 3.9%)	5 (1.1, 3.1%)	10 (2.0, 3.7%)	77 (1.8, 5.9%)
**Antivirals for Systemic Use**	7 (0.5, 2.4%)	12 (2.4, 6.7%)	22 (1.5, 5.4%)	11 (2.5, 6.8%)	13 (2.6, 4.9%)	65 (1.5, 5.0%)
**Blood and Blood Forming Organs**	5 (0.4, 1.7%)	9 (1.8, 5.0%)	28 (1.9, 6.9%)	6 (1.4, 3.7%)	12 (2.4, 4.5%)	60 (1.4, 4.6%)
**Musculo-Skeletal System**	21 (1.6, 7.1%)	5 (1.0, 2.8%)	16 (1.1, 3.9%)	3 (0.7, 1.9%)	10 (2.0, 3.7%)	55 (1.3, 4.2%)
**Dermatologicals**	17 (1.3, 5.7%)	9 (1.8, 5.0%)	11 (0.8, 2.7%)	5 (1.1, 3.1%)	9 (1.8, 3.4%)	51 (1.2, 3.9%)
**Respiratory System**	9 (0.7, 3.0%)	11 (2.2, 6.1%)	13 (0.9, 3.2%)	8 (1.8, 4.9%)	5 (1.0, 1.9%)	46 (1.1, 3.5%)
**Sensory Organs**	7 (0.5, 2.4%)	7 (1.4, 3.9%)	21 (1.4, 5.2%)	3 (0.7, 1.9%)	7 (1.4, 2.6%)	45 (1.1, 3.4%)
**Genito Urinary System and Sex Hormones**	8 (0.6, 2.7%)	3 (0.6, 1.7%)	19 (1.3, 4.7%)	4 (0.9, 2.5%)	5 (1.0, 1.9%)	39 (0.9, 3.0%)
**Antiparasitic Products, Insecticides and Repellents**	10 (0.8, 3.4%)	4 (0.8, 2.2%)	5 (0.3, 1.2%)	1 (0.2, 0.6%)	8 (1.6, 3.0%)	28 (0.7, 2.1%)
**Systemic Hormonal Preparations,** **Excluding Sex Hormones and Insulins**	6 (0.5, 2.0%)	(0.0, 0.0%)	10 (0.7, 2.5%)	2 (0.5, 1.2%)	7 (1.4, 2.6%)	25 (0.6, 1.9%)
**Other Antiinfectives for Systemic Use**	4 (0.3, 1.4%)	1 (0.2, 0.6%)	7 (0.5, 1.7%)	5 (1.1, 3.1%)	2 (0.4, 0.7%)	19 (0.5, 1.5%)
**All Other Therapeutic Products**	5 (0.4, 1.7%)	4 (0.8, 2.2%)	14 (1.0, 3.4%)	4 (0.9, 2.5%)	5 (1.0, 1.9%)	32 (0.8, 2.4%)
**Total**	**296 (23.1, 100%)**	**179 (35.4, 100%)**	**406 (27.7, 100%)**	**162 (36.6, 100%)**	**267 (53.2, 100%)**	**1310 (31.2, 100%)**

Drug approvals by regulatory period, (annual average of approvals, percentage of total approvals).

**Table 2 healthcare-11-01759-t002:** Market discontinuations of FDA-approved drugs by therapeutic class, 1980–2021.

Therapeutic Class	1980–PDUFA (1992)	PDUFA (1992)–FDAMA (1997)	FDAMA (1997)–FDASIA (2012)	FDASIA (2012)–21st Century Cures Act (2016)	21st Century Cures Act (2016)–2021	Total (1981–2021)
**Antineoplastic and Immunomodulating Agents**	7 (0.5, 2.4%)	-	8 (0.5, 2.0%)	2 (0.5, 1.2%)	3 (0.6, 1.1%)	20 (0.5, 1.5%)
**Nervous System**	5 (0.4, 1.7%)	2 (0.4, 1.1%)	2 (0.1, 0.5%)	-	-	9 (0.2, 0.7%)
**Alimentary Tract and Metabolism**	2 (0.2, 0.7%)	3 (0.6, 1.7%)	5 (0.3, 1.2%)	2 (0.5, 1.2%)	-	12 (0.3, 0.9%)
**Cardiovascular System**	12 (0.9, 4.1%)	5 (1.0, 2.8%)	2 (0.1, 0.5%)	1 (0.2, 0.6%)	-	20 (0.5, 1.5%)
**Diagnostic drugs**	14 (1.1, 4.7%)	11 (2.2, 6.1%)	7 (0.5, 1.7%)	-	-	32 (0.8, 2.4%)
**Antibacterials for Systemic Use**	21 (1.6, 7.1%)	4 (0.8, 2.2%)	7 (0.5, 1.7%)	-	-	32 (0.8, 2.4%)
**Antivirals for Systemic Use**	3 (0.2, 1.0%)	5 (1.0, 2.8%)	5 (0.3, 1.2%)	3 (0.7, 1.9%)	-	16 (0.4, 1.2%)
**Blood and Blood Forming Organs**	1 (0.1, 0.3%)	4 (0.8, 2.2%)	5 (0.3, 1.2%)	-	1 (0.2, 0.4%)	11 (0.3, 0.8%)
**Musculo-Skeletal System**	7 (0.5, 2.4%)	2 (0.4, 1.1%)	3 (0.2, 0.7%)	1 (0.2, 0.6%)	-	13 (0.3, 1.0%)
**Dermatologicals**	3 (0.2, 1.0%)	-	3 (0.2, 0.7%)	-	-	6 (0.1, 0.5%)
**Respiratory System**	5 (0.4, 1.7%)	1 (0.2, 0.6%)	2 (0.1, 0.5%)	-	-	8 (0.2, 0.6%)
**Sensory Organs**	2 (0.2, 0.7%)	2 (0.4, 1.1%)	5 (0.3, 1.2%)	2 (0.5, 1.2%)	-	11 (0.3, 0.8%)
**Genito Urinary System and Sex Hormones**	1 (0.1, 0.3%)	-	-	-	-	1 (0.0, 0.1%)
**Antiparasitic Products, Insecticides and Repellents**	5 (0.4, 1.7%)	1 (0.2, 0.6%)	1 (0.1, 0.2%)	-	1 (0.2, 0.4%)	8 (0.2, 0.6%)
**Systemic Hormonal Preparations, Excluding Sex Hormones and Insulins**	3 (0.2, 1.0%)	-	1 (0.1, 0.2%)	-	-	4 (0.1, 0.3%)
**Other Antiinfectives for Systemic Use**	1 (0.1, 0.3%)	-	0 (0.0, 0.0%)	-	-	1 (0.0, 0.1%)
**All Other Therapeutic Products**	2 (0.2, 0.7%)	1 (0.2, 0.6%)	3 (0.2, 0.7%)	-	-	6 (0.1, 0.5%)
**Total**	**94 (7.3, 31.9%)**	**41 (8.1, 22.8%)**	**59 (4.0, 14.5%)**	**11 (2.5, 6.8%)**	**5 (1.0, 1.9%)**	**210 (5.0, 16.0%)**

Drug discontinuations by regulatory period (annual average of discontinuations; discontinuations as percentage of total approvals).

**Table 3 healthcare-11-01759-t003:** FDA approvals, discontinuations, and percentage of market discontinuations of new systemic antibiotics by pharmacological and chemical classes, 1980–2021.

Pharmacological and Chemical Classes	1980–PDUFA (1992)	PDUFA (1992)–FDAMA (1997)	FDAMA (1997)–FDASIA (2012)	FDASIA (2012)–21st Century Cures Act (2016)	21st Century Cures Act (2016)–2021	Total (1981–2021)
**Aminoglycoside Antibacterials**	2 (2, 100.0%)	-	-	-	1 (0, 0.0%)	3 (2, 66.7%)
Other aminoglycosides	2 (2, 100.0%)	-	-	-	1 (0, 0.0%)	3 (2, 66.7%)
**Beta-Lactam Antibacterials, Penicillins**	7 (4, 57.1%)	1 (0, 0.0%)	-	-	-	8 (4, 50.0%)
Combinations of penicillins, incl. beta-lactamase inhibitors	2 (0, 0.0%)	1 (0, 0.0%)	-	-	-	3 (0, 0.0%)
Penicillins with extended spectrum	5 (4, 80.0%)	-	-	-	-	5 (4, 80.0%)
**Macrolides, Lincosamides and Streptogramins**	2 (0, 0.0%)	1 (1, 100.0%)	2 (1, 50.0%)	-	-	5 (2, 40.0%)
Macrolides	2 (0, 0.0%)	1 (1, 100.0%)	1 (1, 100.0%)	-	-	4 (2, 50.0%)
Streptogramins	-	-	1 (0, 0.0%)	-	-	1 (0, 0.0%)
**Other Antibacterials**	-	1 (0, 0.0%)	3 (0, 0.0%)	3 (0, 0.0%)	2 (0, 0.0%)	9 (0, 0.0%)
Glycopeptide antibacterials	-	-	1 (0, 0.0%)	2 (0, 0.0%)	-	3 (0, 0.0%)
Imidazole derivatives	-	-	-	-	1 (0, 0.0%)	1 (0, 0.0%)
Other antibacterials	-	1 (0, 0.0%)	2 (0, 0.0%)	1 (0, 0.0%)	1 (0, 0.0%)	5 (0, 0.0%)
**Other Beta-Lactam Antibacterials**	20 (10, 50.0%)	3 (1, 33.3%)	5 (2, 40.0%)	2 (0, 0.0%)	3 (0, 0.0%)	33 (13, 39.4%)
Carbapenems	1 (0, 0.0%)	1 (0, 0.0%)	2 (1, 50.0%)	-	2 (0, 0.0%)	6 (1, 16.7%)
Fourth-generation cephalosporins	-	1 (0, 0.0%)	-	-	-	1 (0, 0.0%)
Monobactams	1 (0, 0.0%)	-	-	-	-	1 (0, 0.0%)
Other cephalosporins and penems	-	-	1 (0, 0.0%)	1 (0, 0.0%)	1 (0, 0.0%)	3 (0, 0.0%)
Second-generation cephalosporins	8 (5, 62.5%)	-	-	-	-	8 (5, 62.5%)
Third-generation cephalosporins	10 (5, 50.0%)	1 (1, 100.0%)	2 (1, 50.0%)	1 (0, 0.0%)	-	14 (7, 50.0%)
**Quinolone Antibacterials**	7 (5, 71.4%)	2 (2, 100.0%)	5 (4, 80.0%)	-	1 (0, 0.0%)	15 (11, 73.3%)
Fluoroquinolones	6 (4, 66.7%)	2 (2, 100.0%)	5 (4, 80.0%)	-	1 (0, 0.0%)	14 (10, 71.4%)
Other quinolones	1 (1, 100.0%)	-	-	-	-	1 (1, 100.0%)
**Tetracyclines**	-	-	1 (0, 0.0%)	-	3 (0, 0.0%)	4 (0, 0.0%)
Tetracyclines	-	-	1 (0, 0.0%)	-	3 (0, 0.0%)	4 (0, 0.0%)
**Total**	38 (21, 55.3%)	8 (4, 50.0%)	16 (7, 43.8%)	5 (0, 0.0%)	10 (0, 0.0%)	77 (32, 41.6%)

FDA approved systemic antibiotics (annual average of discontinuations; discontinuations as percentage of total approvals).

## Data Availability

Publicly available datasets were analyzed in this study. This data can be found here: Food and Drug Administration (FDA). New Drugs at FDA: CDER’s New Molecular Entities and New Therapeutic Biological Products. https://www.accessdata.fda.gov/scripts/cder/daf/ (accessed on 20 June 2022).

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
