# Peer review of "A Comparative Assessment of Approvals and Discontinuations of Systemic Antibiotics and Other Therapeutic Areas"

_healthcare, 2023, doi:10.3390/healthcare11121759_

Round 1
Reviewer 1 Report
The current study aims to assess the trends in the approvals and discontinuations of systemic antibiotics by the FDA in light of the major regulatory actions that have been implemented chronologically in the US. Overall, the language used is clear and the flow of ideas is good. However, I have a few comments that I would like the authors to address before I can endorse the manuscript for publication.
Introduction
I appreciate the authors' focus on the trends in the approvals and discontinuations of systemic antibiotics by the FDA. However, it would be helpful if the authors could elaborate on why they chose to focus specifically on antibiotics and not other therapeutic classes. Providing this context will help readers better understand the value of the study and its potential implications for clinical practice. Additionally, I would suggest that the authors highlight the specific value that their study will provide, and potential future applications of their findings. This could include discussing how their results might inform future regulatory decision-making processes, or how their findings could be used to improve antibiotic prescribing practices and combat antibiotic resistance. Clarifying these points will enhance the impact of the study and make it more compelling for readers.
Methods
Thank you for providing details for the methods used in the study. However, it is not entirely clear what type of study the authors have conducted. In the methods section, I would suggest that the authors clearly state the type of study. For example, if the study is a content analysis, this should be stated explicitly. In addition, I would recommend that the authors consider adding an introductory chart or diagram in the methods section to provide a clear and concise overview of the methods they used. This could help readers quickly understand the study design and methods, and better appreciate the authors' contributions.
Line 92: “We collected study data using a standardized protocol developed by the authors”. Can the authors please provide more information on how the protocol was standardized? Were any reference protocols used to develop the standardized protocol? Furthermore, I would appreciate clarification on whether this was the first time the protocol has been used, or whether it has been used in previous publications. This information would help readers better understand the validity and reliability of the data collected.
Results
I noticed that the authors use different reporting systems (i.e., N or % or both) in different parts of the results section. To improve the clarity and consistency of their reporting, I suggest that the authors use a consistent reporting system (N, %) throughout this section.
Line 114: While the authors' discussion on the average annual approval rates of other drug classes is informative, I believe it would be more meaningful to provide a discussion of the average annual approval rates specifically for the antibiotic class of drugs, which is the main focus of the study.
Figure 1: I suggest that the authors consider adding the time in years to the columns of the table, to make it easier for readers to track the regulatory periods. Including this information would also help the reader to more easily understand the trends in regulatory actions over time.
Tables 1 and 2: I appreciate the inclusion of Tables 1 &, which provides a detailed breakdown of the antibiotics approved and discontinued during the study period. However, I would like to request clarification on the use of the gray highlighting in the table. Specifically, I noticed that the row for "sensory organs" is highlighted in gray. Can the authors please explain the reason for this highlighting, as it is not immediately clear from the table what this signifies?
Table2: “FDAMA-DASIA”. Please correct this typo.
In the manuscript that the authors state that "All discontinued systemic antibiotics were approved prior to FDASIA in 2012" (line 129). However, upon reviewing Table 2, it appears that most discontinuations occurred prior to 2012, regardless of the drug classification. To avoid potential confusion for the reader, I suggest the authors acknowledge in the results section that most discontinuations occurred prior to the implementation of FDASIA, not just for systemic antibiotics, but for other drug classes as well.
Line 100: Please define the acronym “OB”.
Limitations
I appreciate the authors' efforts to assess the trends of approvals and discontinuations of systemic antibiotics by the FDA over time. However, it is worth noting that the study has a limitation in that it only assessed when drugs were approved, but did not provide information on the market lifetime of the drugs, which could be important for understanding the impact of regulatory actions on the antibiotic market. I suggest that the authors discuss this limitation in their discussion section and consider its potential implications for their findings and conclusions. Additionally, the authors may wish to acknowledge that future research that assesses both approvals and discontinuations of antibiotics could provide a more comprehensive understanding of how regulatory actions have affected the antibiotic market over time.
Author Response
Reviewer 1
Comments and Suggestions for Authors
The current study aims to assess the trends in the approvals and discontinuations of systemic antibiotics by the FDA in light of the major regulatory actions that have been implemented chronologically in the US. Overall, the language used is clear and the flow of ideas is good. However, I have a few comments that I would like the authors to address before I can endorse the manuscript for publication.
Introduction
I appreciate the authors' focus on the trends in the approvals and discontinuations of systemic antibiotics by the FDA. However, it would be helpful if the authors could elaborate on why they chose to focus specifically on antibiotics and not other therapeutic classes. Providing this context will help readers better understand the value of the study and its potential implications for clinical practice. Additionally, I would suggest that the authors highlight the specific value that their study will provide, and potential future applications of their findings. This could include discussing how their results might inform future regulatory decision-making processes, or how their findings could be used to improve antibiotic prescribing practices and combat antibiotic resistance. Clarifying these points will enhance the impact of the study and make it more compelling for readers.
We appreciate the reviewer suggestion and the opportunity to provide some context on the significance of our study. Over the last four decades, the US Congress has passed legislation that offers several incentives to encourage the development and regulatory approval of pharmaceuticals, especially antibiotics (FDAMA, the GAIN Act, the QIDP, the 21st Century Cures Act). Previous studies evaluated antibiotic drug approvals and discontinuations, but they focused on antibiotic approvals within a limited timeframe and did not compare antibiotic approvals with all other therapeutic categories for reference purposes.
As antibiotic resistance poses a significant public health risk and concerns grow over potential multi-resistant bacterial pathogens, there is an urgent need for novel antibiotics and research in this area.(1,2) Therefore, this study aims to shed light on the effectiveness of US regulatory measures and related policies in promoting the development of antibiotics in general and novel antibiotics addressing unmet patient needs in particular. Study findings may inform legal and regulatory systems as well as public health programs and advance understanding of the effectiveness of enacted laws and implemented policies bringing more and novel systemic antibiotics to the US market compared to available therapeutic alternatives. Please see study rationale provided in the Introduction section of the manuscript.
Methods
Thank you for providing details for the methods used in the study. However, it is not entirely clear what type of study the authors have conducted. In the methods section, I would suggest that the authors clearly state the type of study. For example, if the study is a content analysis, this should be stated explicitly. In addition, I would recommend that the authors consider adding an introductory chart or diagram in the methods section to provide a clear and concise overview of the methods they used. This could help readers quickly understand the study design and methods, and better appreciate the authors' contributions.
Thank you. We added a statement on the study design in the methods section. This was an observational (retrospective) study using publicly available data.
Line 92: “We collected study data using a standardized protocol developed by the authors”. Can the authors please provide more information on how the protocol was standardized? Were any reference protocols used to develop the standardized protocol? Furthermore, I would appreciate clarification on whether this was the first time the protocol has been used, or whether it has been used in previous publications. This information would help readers better understand the validity and reliability of the data collected.
Per the reviewer suggestion, we added a clarifying statement. We collected study data using a standardized data extraction form developed and piloted by the authors and used in prior studies.
Results
I noticed that the authors use different reporting systems (i.e., N or % or both) in different parts of the results section. To improve the clarity and consistency of their reporting, I suggest that the authors use a consistent reporting system (N, %) throughout this section.
Thank you. We thoroughly reviewed the results section and consistently reported both N and %.
Line 114: While the authors' discussion on the average annual approval rates of other drug classes is informative, I believe it would be more meaningful to provide a discussion of the average annual approval rates specifically for the antibiotic class of drugs, which is the main focus of the study.
Thank you for the suggestion. We agree with the reviewer and have focused discussion on systemic antibiotics. However, for the purpose of supporting our arguments and providing context, we have referred to other therapeutic classes in a few instances in the discussion section. For example, it is noteworthy that antibiotics for systemic use ranked sixth in number of approvals but first in market discontinuations. Our study findings show that market discontinuations of systemic antibiotics occurred at a rate three times higher than that of other drug classes. Another significant finding is that, with regard to the disincentive argument for developing new anti-infectives for acute vs. chronic drug use, we observed a similar downward approval trend in other therapeutic classes indicated for prevalent and chronic conditions, such as cardiovascular drugs.
Figure 1: I suggest that the authors consider adding the time in years to the columns of the table, to make it easier for readers to track the regulatory periods. Including this information would also help the reader to more easily understand the trends in regulatory actions over time.
Thanks. We added year of enactment both to Figure 1 and Tables
Tables 1 and 2: I appreciate the inclusion of Tables 1 &, which provides a detailed breakdown of the antibiotics approved and discontinued during the study period. However, I would like to request clarification on the use of the gray highlighting in the table. Specifically, I noticed that the row for "sensory organs" is highlighted in gray. Can the authors please explain the reason for this highlighting, as it is not immediately clear from the table what this signifies?
Thank you for the opportunity to clarify. The highlighted rows serve no other purpose than to facilitate reading of the table by providing visual breaks. Revised tables do not include any highlighted rows.
Table 2: “FDAMA-DASIA”. Please correct this typo.
Thank you for pointing out this typo. It reads now “FDAMA-FDASIA”.
In the manuscript that the authors state that "All discontinued systemic antibiotics were approved prior to FDASIA in 2012" (line 129). However, upon reviewing Table 2, it appears that most discontinuations occurred prior to 2012, regardless of the drug classification. To avoid potential confusion for the reader, I suggest the authors acknowledge in the results section that most discontinuations occurred prior to the implementation of FDASIA, not just for systemic antibiotics, but for other drug classes as well.
Thank you for the opportunity to clarify. We stated in the manuscript that drug market discontinuations are likely correlated with the number of years after market entry. Hence, the overall trend in drug discontinuations pointed out by the reviewer is accurate. What it is worth noting is the discontinuation rate of antibiotics compared to other therapeutic areas. Market discontinuations of systemic antibiotics occurred at three times the discontinuation rate in other drug classes. The Pearson’s chi-squared test for independence confirmed this finding – see below.
Line 100: Please define the acronym “OB”.
Thank you. This was an oversight. OB stands for Orange Book
Limitations
I appreciate the authors' efforts to assess the trends of approvals and discontinuations of systemic antibiotics by the FDA over time. However, it is worth noting that the study has a limitation in that it only assessed when drugs were approved but did not provide information on the market lifetime of the drugs, which could be important for understanding the impact of regulatory actions on the antibiotic market. I suggest that the authors discuss this limitation in their discussion section and consider its potential implications for their findings and conclusions. Additionally, the authors may wish to acknowledge that future research that assesses both approvals and discontinuations of antibiotics could provide a more comprehensive understanding of how regulatory actions have affected the antibiotic market over time.
Thanks. We acknowledged in the manuscript, and mentioned in our response to the prior question, that drug market discontinuations may be likely correlated with the market lifetime of drugs across therapeutic classes not only systemic antibiotics. We stated in the limitations that study findings need to be interpreted in the context of regulatory changes implemented during the study period and the number of years an antibiotic has been in the market. Yet, the antibiotic discontinuation rate was three times the discontinuation rate of other therapeutic classes during the study period.
References
(1) Jernigan JA, Hatfield KM, Wolford H, Nelson RE, Olubajo B, Reddy SC, McCarthy N, Paul P, McDonald LC, Kallen A, Fiore A, Craig M, Baggs J. Multidrug-Resistant Bacterial Infections in U.S. Hospitalized Patients, 2012-2017. N Engl J Med. 2020;382(14):1309-1319. doi: 10.1056/NEJMoa1914433. PMID: 32242356.
(2) Antimicrobial Resistance Collaborators. Global burden of bacterial antimicrobial resistance in 2019: A systematic analysis. Lancet. 2022;399(10325):629-655. doi: 10.1016/S0140-6736(21)02724-0. Epub 2022 Jan 19. Erratum in: Lancet. 2022 Oct 1;400(10358):1102. PMID: 35065702.
Reviewer 2 Report
This is a study of FDA-approved drugs over a period of time. focusing on antibiotics and comparing them with other classes of drugs. There are few antibiotics that are approved, and each time the bacterial resistance is greater. We are certainly facing a challenge. I would like to know if the authors consider that their article can be a starting point for new research.
Your work focuses on a study on FDA approvals. The appearance of new drugs as a measure of study. Undoubtedly we find ourselves with a problem with the appearance of resistance to antibiotics.
I would like to know if the authors consider that their article can be a starting point for new research.
I believe that including other drug agencies such as the EMA can contribute
Author Response
Reviewer 2
Comments and Suggestions for Authors
This is a study of FDA-approved drugs over a period of time, focusing on antibiotics and comparing them with other classes of drugs. There are few antibiotics that are approved, and each time the bacterial resistance is greater. We are certainly facing a challenge. I would like to know if the authors consider that their article can be a starting point for new research. Your work focuses on a study on FDA approvals. The appearance of new drugs as a measure of study. Undoubtedly, we find ourselves with a problem with the appearance of resistance to antibiotics. I believe that including other drug agencies such as the EMA can contribute.
Thank you; This is a great idea. We added following statement to the manuscript and plan to conduct the FDA and EMA comparative analysis. Future research could compare the regulatory approval pathways, designations, and indications of systemic antibiotics between the FDA and the European Medicines Agency.
Reviewer 3 Report
The work entitled “A Comparative Assessment of Approvals and Discontinuations 2 of Systemic Antibiotics and Other Therapeutic Areas” by Rosa Rodriguez-Monguio and collaborators compares the medians of different drug classes approved by FDA and discontinued drugs in different periods of times.
The work is well written, but some limitations are found in the description of their work that does not allow the reader to establish the relevance of their claims. In particular:
Major concerns
1. The authors do not consider that in their comparisons the data comprises different number of years, and presumably different sample sizes (different number of drugs). For instance, the period from 1980-PDUFA (1992) includes 12 years, while the period PDUFA-FDAMA includes only 5 years (1992-1997). This generates a bias in their data that the authors are not considering in their analysis. To conclude that the differences observed in the reported media are significant, the authors need to run statistical tests. For instance, the authors may use the one-way ANOVA test if their samples can be shown to distribute normally; otherwise, the authors may use the Mann-Whitney U-test. It is advisable for the authors to seek statistical advice.
2. The description of their methods is poor. It is not possible for a reader to understand exactly what information was obtained and how the collected data was grouped according to the reported classes in their figures and tables. It is noticeable that only 1 antimicrobial peptide is included in their data; this suggests that the data is not complete. By looking at FDA-approved compounds from the ZINC database (http://zinc.docking.org/substances/subsets/fda/), the total number of listed compounds is 1310 approved compounds and 208 discontinued compounds, which are similar numbers than those reported in this work. On the other hand, the PubChem includes 4348 compounds from the FDA Orange Book drugs (identified by their ID numbers, CID), which includes all FDA approved drugs (https://pubchem.ncbi.nlm.nih.gov/source/11932). PubChem provides the information authors need (see for instance https://pubchem.ncbi.nlm.nih.gov/docs/pug-rest), hence it may constitute a better source to answer the questions relevant to this work. It is possible that this information may provide authors with additional tools to perform their analysis, but more importantly, to improve the description of their method, for instance to use compound unique identifiers in addition to compound names (which are many times ambiguos), among others.
3. The authors speculate too much in the discussion of their results. Since the data presented can only show differences, if any, between approval and discontinuation of different drug classes, it would be desirable that authors would discuss implications of such differences and support these implications based on previous published works. For instance, if a drug class (e.g., cardiovascular system) has a larger approval rate than discontinuation, this suggests that these drugs are well tested and/or have stable markets; which one of these possibilities is supported by other findings in the literature?
4. From the explanatory text about each FDA legislation about approved drugs, it seems none of these legislations would affect the discontinuation rate. If so, it would be convenient for authors to show the discontinuation averages per year for the different periods compared in the approvals. Alternatively, it is advisable to show the distribution of number of discontinued drugs per year to test for any bias.
Minor concerns
1. Figure 1 and tables, the names PDUFA-FDAMA, FDAMA-FDASIA, FDASIA 21st CCA and 21st CCA-2021 do not include the year of each program. It will help to include those years so that the reader may be aware that different times are boing compared.
2. In the text, the reader could guess that the abbreviation used in the figures and tables ,21st CCT, refers to 21st century cures act approved in 2016, but the abbreviation was not defined.
3. Line 100, it states that “…it was listed in the OB discontinuation section…”, OB is not defined; is this the Orange Book?
4. Table 2 does not specify, as table 1 does, what percentages are ported.
5. Line 162 “staphylococcus aureus” should be Staphylococcus aureus in italics.
6. Some of the links provided in the reference section do not link to any page; for instance, reference 7, 10, 11 and 16.
Author Response
Reviewer 3
Comments and Suggestions for Authors
The work entitled “A Comparative Assessment of Approvals and Discontinuations 2 of Systemic Antibiotics and Other Therapeutic Areas” by Rosa Rodriguez-Monguio and collaborators compares the medians of different drug classes approved by FDA and discontinued drugs in different periods of times.
The work is well written, but some limitations are found in the description of their work that does not allow the reader to establish the relevance of their claims. In particular:
Major concerns
- The authors do not consider that in their comparisons the data comprises different number of years, and presumably different sample sizes (different number of drugs). For instance, the period from 1980-PDUFA (1992) includes 12 years, while the period PDUFA-FDAMA includes only 5 years (1992-1997). This generates a bias in their data that the authors are not considering in their analysis. To conclude that the differences observed in the reported media are significant, the authors need to run statistical tests. For instance, the authors may use the one-way ANOVA test if their samples can be shown to distribute normally; otherwise, the authors may use the Mann-Whitney U-test. It is advisable for the authors to seek statistical advice.
Per the reviewer suggestion, we acknowledged that the spam of the regulatory periods is not equal and there may be an accumulative effect as those regulations and policies were sequentially implemented. To the reviewer’s point on drug selection bias, there is not such a risk of bias because our study includes the entire population of systemic antibiotics approved by the FDA since 1980. We used descriptive statistics to characterize the study sample. To assess statistically significant differences in discontinuations by therapeutic class, we used non-parametric Pearson’s chi-squared test of independence and reported the p-values.
- The description of their methods is poor. It is not possible for a reader to understand exactly what information was obtained and how the collected data was grouped according to the reported classes in their figures and tables. It is noticeable that only 1 antimicrobial peptide is included in their data; this suggests that the data is not complete. By looking at FDA-approved compounds from the ZINC database (http://zinc.docking.org/substances/subsets/fda/), the total number of listed compounds is 1310 approved compounds and 208 discontinued compounds, which are similar numbers than those reported in this work. On the other hand, the PubChem includes 4348 compounds from the FDA Orange Book drugs (identified by their ID numbers, CID), which includes all FDA approved drugs (https://pubchem.ncbi.nlm.nih.gov/source/11932). PubChem provides the information authors need (see for instance https://pubchem.ncbi.nlm.nih.gov/docs/pug-rest), hence it may constitute a better source to answer the questions relevant to this work. It is possible that this information may provide authors with additional tools to perform their analysis, but more importantly, to improve the description of their method, for instance to use compound unique identifiers in addition to compound names (which are many times ambiguous), among others.
Thank you for the suggestion and the opportunity to clarify. We have expanded the methods section to provide a more detailed account of the data collection process. In response to the reviewer's suggestion to utilize the active ingredient as the unit of analysis, the same active ingredient may have multiple strengths, administration routes, and other characteristics. As a result, utilizing the active ingredient alone would not offer the level of granularity necessary for the purposes of this study.
The dataset is comprehensive and encompasses the entire population of systemic antibiotics approved by the FDA since 1980. However, we acknowledged in our limitations section that our study focuses solely on systemic antibiotics and excludes topical antibiotics. Furthermore, as noted in the manuscript, our analysis is limited to the first approval of systemic antibiotics. We included all new molecular entities, drugs containing an active ingredient that has never been previously approved for marketing in the US, approved by the FDA since 1980. We adopted the first new drug application or biologic license application for each new molecular entity as the unit of analysis. To identify systemic antibiotics, we utilized the unique identifier National Drug Code (NDC). The NDC serves to uniquely identify the drug manufacturer, active ingredient, route, form, strength, and packaging. The use of the NDC is the standard practice when conducting these analyses.
- The authors speculate too much in the discussion of their results. Since the data presented can only show differences, if any, between approval and discontinuation of different drug classes, it would be desirable that authors would discuss implications of such differences and support these implications based on previous published works. For instance, if a drug class (e.g., cardiovascular system) has a larger approval rate than discontinuation, this suggests that these drugs are well tested and/or have stable markets; which one of these possibilities is supported by other findings in the literature?
To address the reviewer comment we acknowledged in the manuscript that therapeutic classes with the greatest number of approvals were antineoplastic, central nervous system, and alimentary tract and metabolism drugs, and cardiovascular. Conversely, the drug classes with the greatest number of discontinuations were antibacterials for systemic use, diagnostic drugs, and antivirals. These findings support the trends identified in earlier studies conducted over shorter timeframes and previous periods [5,6].
As suggested by the reviewer, it is plausible that drug classes with more robust evidence of efficacy and safety available at the time of regulatory approval such as nervous system and cardiovascular system drugs indicated for chronic conditions may exhibit lower discontinuation rates. However, although these explanations are plausible, we cannot determine the underlying causes of drug discontinuations beyond the safety withdrawals evaluated in this study. We stated in the discussion that the short-term treatment course, compared to drugs indicated for chronic conditions, has been pointed as the reason behind the high market discontinuation rate. Nevertheless, in this study, and others before [5], we found an overall downward trend in the approval of systemic antibiotics comparable to the trend observed for drugs for chronic, prevalent conditions including cardiovascular and musculo-skeletal system drugs.
- From the explanatory text about each FDA legislation about approved drugs, it seems none of these legislations would affect the discontinuation rate. If so, it would be convenient for authors to show the discontinuation averages per year for the different periods compared in the approvals. Alternatively, it is advisable to show the distribution of number of discontinued drugs per year to test for any bias.
Per reviewer suggestion, we included number of discontinuations by therapeutic class and regulatory period, annual average of discontinuations, and discontinuations as percentage of approvals in Table 2.
Minor concerns
- Figure 1 and tables, the names PDUFA-FDAMA, FDAMA-FDASIA, FDASIA 21st CCA and 21st CCA-2021 do not include the year of each program. It will help to include those years so that the reader may be aware that different times are boing compared.
We included years to the enactment of each law in the tables.
- In the text, the reader could guess that the abbreviation used in the figures and tables ,21st CCT, refers to 21st century cures act approved in 2016, but the abbreviation was not defined.
We have spelled out 21st Century Cures Act in Table 1
- Line 100, it states that “…it was listed in the OB discontinuation section…”, OB is not defined; is this the Orange Book?
We have spelled out Orange Book (OB) the first time it appears in the text
- Table 2 does not specify, as table 1 does, what percentages are ported.
We included years to the enactment of each law in the tables. - Line 162 “staphylococcus aureus” should be Staphylococcus aureus in italics.
Corrected. Thanks.
- Some of the links provided in the reference section do not link to any page; for instance, reference 7, 10, 11 and 16.
Thank you. We reviewed and updated links included in the references
Reviewer 4 Report
The article analyzes trends and I think there is merit for publication.
There are observations regarding the analysis of the data and the quality of the graphics needs to be improved

Author Response
Reviewer 4 –
This study assessed long-term trends and characteristics of FDA approved new systemic antibiotics and discontinuations, evaluated reasons leading to those market discontinuations, and compared systemic antibiotic approvals and discontinuations with all other therapeutic classes in the context of laws enacted and regulations implemented in the US over the last forty years.
General observations:
- As a suggestion, include in the methods section that the study is a narrative review.
Thank you. We added a statement on the study design in the methods section. This was an observational (retrospective) study using publicly available data.
- As a suggestion, describe in more detail the data sources used in the review. Was the data obtained directly from the website? Was the data solicited from information sources? this suggestion can make the study wider for the external reader
Thank you for the suggestion. We expanded the Materials and Methods section and provided more details on the information collected from each data sourced.
- Tables 1 and 2. I suggest to include in the tables additional columns with the absolute and percentage variance
Per reviewer suggestion, we included number of approvals by therapeutic class and regulatory period, annual average of approvals, and as percentage of total approvals in Table 1. Similarly, we included the number of discontinuations by therapeutic class and regulatory period, annual average of discontinuations, and discontinuations as percentage of approvals in Table 2.
Round 2
Reviewer 1 Report
I would like to thank the authors for replying to the comments and revising the manuscript. However, I still believe the authors should indicate in the limitations that the study does not provide the market live time of the antibiotics. The authors state in the last paragraph of the discussion that “Study findings need to be interpreted in the context of regulatory changes implemented during the study period and the number of years an antibiotic has been in the market”. However, it is clear that the study does not provide the “number of years an antibiotic has been in the market”. This should be modified and highlighted in the limitations section.
Author Response
We would like to kindly draw the reviewer's attention to the fact that the statement regarding the number of years an antibiotic has been on the market was included as a part of the study limitations. We have copied the paragraph below for the reviewer's reference.
“This study has some limitations. …. Study findings need to be interpreted in the context of regulatory changes implemented during the study period and the number of years an antibiotic has been in the market…”
Reviewer 2 Report
The work has improved, as an annotation to make figure 1 and figure 3 the letters should be seen better
Author Response
Thank you. We re-did the figures and increased the font size.
Reviewer 3 Report
The work entitled “A Comparative Assessment of Approvals and Discontinuations 2 of Systemic Antibiotics and Other Therapeutic Areas” by Rosa Rodriguez-Monguio and collaborators compares the medians of different drug classes approved by FDA and discontinued drugs in different periods of times.
The work has been reviewed once and the authors addressed all the previous concerns. However, once the authors clarify some aspects of their work, a major concern is now evident that requires the authors attention to improve the description of their work.
Major concern
1. The author have listed the features extracted for each antibiotic included in this analysis: i) biologic license application, ii) filling and approval dates, iii) product number, and iv) type of application, v) active ingredient, vi) generic and brand names, vii) dosage form and viii) administration route, ix) National Drug Code, x) market status, FDA approval pathway xi) priority review or xii) standard review as well as xiii) orphan drug designation. As stated by the authors, the goal of this study is to identify “…reasons for discontinuations by therapeutic class in the context of laws and regulations implemented over the past four decades…”, hence it is fundamental for authors to describe the relationship of these features with regulation changes. For instance, the FDAMA “encourage development of new systemic antibiotics”; how the extracted features are related to the spirit of the this or any other law considered in this study? In other words, if the goal of this study is to establish reasons of antibiotic discontinuation associated to changes in laws, the study needs to measure features that are related to the law, hence any changes on those features in the rate of discontinuation on a given antibiotic reflects changes in the law. All the presented data include the rate of approvals/discontinuations of antibiotics over time; while it is plausible that some discontinuations after a new law has been approved may be due to changes in that new law, not all discontinuations would be related to changes in that new law. Hence, the extracted features for each antibiotic may help to clarify this aspect. However, no analysis on those features is presented. If there is no obvious relationship between the measured features and the changes on legislation, the reported relationships may be considered spurious. Spurious relationships may be useful to make predictions, yet these may not be used to claim causal relationships. The current description of their work seems to claim there is a causal relationship between the features analyzed and the changes in laws (“…evaluated reasons leading to those market discontinuations…”); without a clear description of the relationship of such features and each law considered in this study, the study cannot claim causal relationships. Thus, it is advisable for authors to re-write their work to either a) clarify this relationship may not be causal and discuss which features may be desirable to have to establish a causal relationship, or b) specify the relationship between the extracted features for each antibiotic with law changes and show changes on those features with time and law changes.
Author Response
Thank you for acknowledging that the authors have addressed all the previous concerns. As per the reviewer's suggestion, we have rewritten the statement from "...evaluated reasons leading to those market discontinuations..." to "...evaluated reasons associated with those market discontinuations...". We have thoroughly reviewed the manuscript and replaced "causal" wording with "association" wording. Additionally, we kindly refer the reviewer to our discussion where we explicitly stated that drug market discontinuations are correlated with the number of years after market entry.